# Violence Risk Assessment and Risk Management: Case-Study of Filicide in an Italian Woman

**DOI:** 10.3390/ijerph19126967

**Published:** 2022-06-07

**Authors:** Antonia Sorge, Giovanni Borrelli, Emanuela Saita, Raffaella Perrella

**Affiliations:** 1Department of Psychology, Catholic University of the Sacred Heart, 20123 Milan, Italy; emanuela.saita@unicatt.it; 2Department of Psychology, University of Campania Luigi Vanvitelli, 81100 Caserta, Italy; giovanni.borrelli.1991@virgilio.it (G.B.); raffaella.perrella@unicampania.it (R.P.)

**Keywords:** infanticide, filicide, violent behaviour, COVID-19, risk-assessment, LS/CMI, HCR-20 V3, woman offender, forensic psychology

## Abstract

Background: At an international level, the risk assessment and management process of violent offenders follows a standard method that implies well-defined theoretical models and the use of scientifically validated tools. In Italy, this process is still highly discretionary. The aim of this study is to highlight the advantages deriving from the use of risk assessment tools within the framework of a single case study; Methods: Recidivism risk and social dangerousness of an Italian woman perpetrator of filicide were assessed through the administration of the Level of Service/Case Management Inventory (LS/CMI) instrument supported by Historical Clinical Risk-20 Version 3 (HCR-20 V3); Results: The administration of LS/CMI showed that, in this single case, the subcomponents represent a criminogenic risk/need factor are: Family/Marital, Companions, Alcohol and Drug Problem and Leisure; while constituting strengths: employment and the absence of a Pro-criminal Orientation and an Antisocial Pattern; Conclusions: Data collected through LS/CMI indicated life areas of a single case, which should be emphasised not only to assess the risk of re-offending and social dangerousness but also for a social rehabilitation programme more suited to the subject. This study demonstrates that the LS/CMI assessment tool is suitable for the Italian context.

## 1. Introduction

During the last two years of the COVID-19 pandemic, domestic violence crimes against women and children have increased. This increase is linked to both isolation stress and children in the home due to school closures [1,2,3,4,5,6]. Moreover, Mamun and colleagues highlighted how economic distress and fear of being infected are considered risk factors for the infanticide–suicide phenomenon [7]. 

According to the National Centre for Injury Prevention and Control, infanticide is the fifth leading cause of death for children under the age of five. Nine times out of ten, it is the mother who kills her child(ren) [8]. 

The United Nations Office on Drugs and Crime (UNODC) [9] estimated that during the ten-year period 2008–2017, a total of 205,153 children aged 0 to 14 years lost their lives worldwide (41 countries included) as a result of homicide. In Italy, 447 children were murdered by their parents during the period 2000–2017 [10].

The literature review showed that most studies report that filicides occur predominantly by the fourth month of a child’s life, and the characteristics of homicidal mothers include young age, low education level, having more than one child, low socioeconomic status, and not having adequate prenatal care knowledge. Relationships with partners and biological fathers are often unstable, dysfunctional, and violent. Most of these women do not experience adequate support systems. The co-occurrence of these factors among young mothers can lead to unpreparedness for the increasing emotional, material, and financial care required by a newborn child [11]. 

Regarding risk factors, some studies agree that filicide is primarily linked to psychosocial stress [12,13,14,15]. For example, trauma, substance abuse [16], precarious financial conditions, relationships marked by discord and violence, a history of childhood abuse, and parental separation are recurrent factors in women who kill their children. Other studies have highlighted the relationship between filicide and society’s unrealistic expectations of motherhood [17]. In one study [18], it was found that 72% of the women who had committed filicide had no specific diagnosis of a mental disorder before the crime. Only four out of 96 women presented with a diagnosis of post-partum depression [11]. Aggressive thoughts and—in some cases—even homicidal ideation towards their children are also found among women in the general population [19]. This especially occurs in cases of intestinal colic and inconsolable crying, where the aim is silencing their baby, even in violent ways [20]. The period between birth and the first year of a child’s life represents a unique moment of biological change that can cause a psychological disturbance. Fifty to eighty percent of women develop the ‘baby blues’ within 4–5 days post-delivery [21]. Symptoms of such conditions include anxiety, unmotivated crying, impatience, irritability, lack of self-confidence, and restlessness [22].

Filicide is highly correlated with postpartum psychosis (PP), which has an incidence of approximately 1–4 cases per 1000 births. However, PP may not be detected by professionals because the symptoms are intermittent, and some women tend to hide them to not disappoint their families [23]. These symptoms are often associated with suicidal ideation, which can take the form of suicide/filicide situations. Personality disorders are also highly correlated with violent crimes: many filicidal mothers have had negative and traumatic experiences in childhood and adolescence, which has impacted their personality development. Anxiety and dependency traits and low self-esteem were particularly observed. These results indicate a complex picture involving social, relational, emotional, and personality aspects. 

Recidivism risk assessment is fundamental in cases of maternal filicide, especially when the mother has other children. Structured interviews can be an important source of data both to formulate a diagnosis of mental pathology and to contextualise the event as much as possible [11]. In Anglo-Saxon countries, the use of scientific tools is customary in many areas—for example, during the eligibility assessment for parole and probation, in psychiatric and forensic hospitals, prisons, courts, and to manage treatment programmes. 

For years, forensic psychology research has been concerned with demonstrating the reliability and predictive validity of violence risk assessment tools [24,25,26,27,28,29,30]. Currently, there are more than 200 violence risk assessment tools with a wide range of applications and varying levels of accuracy [31,32].

In 34 USA states, even the use of validated assessment tools is mandatory for the risk assessment of youth probationers [33,34]. They help professionals assess, manage, and reduce recidivism risk. Risk assessment tools assume even more relevance in adolescents and young adults because of the impact that evaluations and programmes can have on their development.

In Italy, evaluations are carried out by professionals appointed by judges. Although several authors have proposed standardised evaluation models—considering individual, social, and environmental characteristics linked to two macro-categories of indicators (internal and external) [24,35]—the choice of tools to be used is still discretionary. A purely clinical approach leads to subjective and non-generalisable evaluations, conditioned by professional experience. Therefore, the likelihood of predicting the risk of violence or criminal behaviours decreases significantly. In this way, the predictive accuracy of the assessments demonstrated by previous research is no better than chance (see, for example, [24,25]).

The use of standardised tools is also functional to the risk management process, which allows one to identify and monitor the most useful treatment for the offender and to effectively contribute to the prevention and reduction of violence and other forms of reoffending. In addition, the use of such tools promotes greater adherence by professionals to the risk-need-responsivity model (RNR) [36,37,38]. This model comes from the general personality and cognitive social learning theory of criminal conduct [38], and it is currently considered the leading assessment and treatment model for offenders. Based on the RNR model, the Level of Service (LS) instruments allow one to assess the offenders’ recidivism risk considering static risk factors, criminogenic needs, responsiveness factors, as well as their strengths. 

There is an ongoing debate about the sensitivity of violence risk assessment tools for female populations. Their effectiveness is relevant because of the impact of violent behaviours on their children [39]. According to Bonta and Andrews [38], risk assessment tools that allow both assessment and case management are the most suitable. De Vogel and colleagues [40] found that the Level of Supervision Inventory (LSI) [41] is the most effective tool for assessing general and violent recidivism in both men and women. Other studies demonstrated the predictive validity of both Historical Clinical Risk-20 (HCR-20) [42] and Level of Service/Case Management Inventory (LS/CMI) [43] among adult offenders and Youth Level of Service/Case Management Inventory (YLS/CMI) [44] and Structured Assessment of Violence Risk in Youth (SAVRY) [45] among juvenile offenders [46,47,48].

Despite these results, in the forensic field, the most widely used standardised assessment instruments are the Minnesota Multiphasic Personality Inventory (MMPI) [49] and the Rorschach [50]. They are used in 96% and 36% of cases, respectively, against 11% associated with the use of the Psychopathy Checklist-Revised (PCL-R) [51] and 1% of the Violence Risk Appraisal Guide (VRAG) [52,53,54]. MMPI and Rorschach provide useful elements for evaluation, but they do not allow for an objective prediction of violent behaviours.

This research is part of a larger study on the Italian adaptation of the LS/CMI tool. The aim of the present study (see Figure A1) is to assess the re-offending risk of an Italian woman perpetrator of filicide, to give an example of a violence risk assessment procedure based on the use of a risk assessment tool already used worldwide evaluating its suitability in the Italian context.

The objective risk assessment and current social dangerousness of this single case, using the LS/CMI and the HCR-20 V3 instruments, will be useful for the identification and management of social reintegration programmes that respect women’s needs. Moreover, this study represents a first step towards adapting the national evaluation processes to the international panorama in the field of forensic psychology.

## 2. Materials and Methods

### 2.1. Tools

#### 2.1.1. Level of Service/Case Management Inventory (LS/CMI)

The Level of Service/Case Management Inventory (LS/CMI) applies to offenders over 17 years old. The interview guide allows one to assess and measure risk factors, criminogenic needs, and degree of responsiveness of the offender, but it is also a comprehensive case management tool [55]. LS/CMI consists of 11 sections, but only the first one allows identification of the offender’s recidivism risk level. The first section is composed of eight sub-components (criminal history, education/employment, family/marital, leisure/recreation, companions, alcohol/drug problem, pro-criminal attitude/orientation, antisocial pattern), corresponding to static risk factors, and criminogenic needs assessed through 43 items. LS/CMI’s validity and reliability have been tested within various contexts and jurisdictions [56]. Although the Italian validation study of LS/CMI is still ongoing, for this single case study, the Italian version of the LS/CMI—approved by the authors and the publisher Multi-Health Systems (Inc MHS, 2019)—was administered.

#### 2.1.2. Historical Clinical Risk—20 V3 (HCR-20 V3)

The Historical Clinical Risk—20 Version 3 (HCR-20 V3) [57]; Italian ed. by Caretti and colleagues [58] is a risk assessment tool for violent behaviour targeting both offenders and forensic patients over 18 years old. It includes 20 items divided into historical (*n* = 10), clinical (*n* = 5), and future risk management (*n* = 5). In the Italian context, it represents a useful assessment tool that allows for the definition and management of treatment programmes for offenders under detention and/or non-custodial security measures.

### 2.2. Procedure

The current study is a single-case study of an Italian woman convicted of filicide. We chose this research design to evaluate the suitability of LS/CMI in the Italian context, examining a single case that may lead to the use of risk assessment tools in larger populations of Italian violent offenders, facilitating forensic decisions.

Although the Italian version of LS/CMI has not yet been validated, its translation (which had to comply with a very stringent back translation procedure carried out by three native English-speaking professors and psychologists) has been formally accepted by the author and by the publisher Multi-Health Systems Inc. (MHS). The method refers to the risk-need-responsivity model, which is currently considered the most effective method for assessing and managing the risk for criminal behaviour and recidivism by different judicial systems. It derives from general personality and cognitive social learning theory [37].

To scientifically support the risk assessment procedure, we combined the LS/CMI tool with the use of HCR-20 V3, already validated in Italian [58]. HCR-20 V3 is the latest version of a comprehensive set of professional guidelines for violence risk assessment and management based on the Structured Professional Judgement (SPJ) model. It is widely used within the conditional release context.

The choice to examine the case of a filicidal woman was due to the increasing trend of domestic violence observed during the COVID-19 pandemic. Moreover, this case is worthy of scientific interest. In fact, despite her psychiatric history—widely associated with filicide in the literature—our subject has shown considerable improvement over time. For this reason, her supervising judge authorised the current recidivism risk assessment to allow more autonomy in her social reintegration programme.

For this single case, the LS/CMI and HCR-20 V3 were used to assess which factors influence the risk of reoffending and social dangerousness and to identify and manage the most suitable reintegration programme. First, relevant information collected over the years by the various professionals involved in the case are presented (see Box 0), and then data collected through the administration of the tools are analysed.

No written statement of informed consent was taken from the patient because both the evaluation and authorisation to analyse her clinical record and then disclose our scientific results were formally allowed from the patient’s supervising judge and by the head of the therapeutic residential community where she still lives. The assessment procedure took place in the therapeutic community and was carried out by her psychologist after appropriate training in the use of the LS/CMI interview guide. To protect the patient’s privacy, any sensitive data are omitted. In this way, her identity cannot be traced.

The study was approved by the Ethics Commission for Research in Psychology (CERPS) at the Catholic University of the Sacred Heart of Milan, protocol no. 10–19.

## 3. Results

Data collected by administering the LS/CMI’s first section and HCR-20 V3 are presented below. During the assessment phase of this study, Anna was on probation, and she was living in the therapeutic residential community.

### 3.1. LS/CMI Section 1: General Risk/Need Factors

#### 3.1.1. Criminal History (CH) Subcomponent of LS/CMI

Anna was convicted for the first time at the age of 24 for property offences (invasion of grounds and premises). At the age of 32, she was convicted of filicide and placed first in a forensic hospital and then on probation in a therapeutic residential community. During detention, she was never punished for misconduct, never received behavioural reports or suspensions, or revocations of the measures applied to her. Data collected—confirmed by documentary controls—allows the attribution of the following scores for the CH subcomponent items (see Table 1).

Comparing the CH total score with the risk/need levels defined by the LS/CMI authors [40] (see Table 2), we can say that this sub-component represents a low-level risk factor for Anna.

#### 3.1.2. Education/Employment (EE) Subcomponent of LS/CMI

Anna completed her middle school diploma and dropped out of school after her second year of high school. She currently has a permanent job that keeps her busy for 21.5 h per week and has had this job at the therapeutic residential community where she lives for the last four years. Relationships with colleagues and her employer are good. She has never been criticised and has never disturbed other employees, and she willingly follows the instructions of her boss. However, she reports that she is not satisfied with her job, as she feels as if she is not working to her full potential and that she could do more demanding work than her current job. Nevertheless, she appreciates the opportunity to earn money. Based on these data, we attributed the following scores to the EE subcomponent items (see Table 3).

Comparing the total score of EE with the risk/need levels defined by the authors of LS/CMI (see Table 2), we can say that education/employment represents a very low-level risk factor for Anna.

#### 3.1.3. Family/Marital (FM) Subcomponent of LS/CMI

Concerning the relational sphere, Anna has been married for 21 years and has never experienced unfaithfulness, but she had many unwanted pregnancies and voluntary terminations of pregnancies. She reports marital and sexual satisfaction. Anna states that she had disputes with her husband about their children’s upbringing, financial problems, and exes. They never disputed about their friends and how to spend their free time. They never thought of separating or divorcing. She reports no difficulties in communication, openness, warmth, or intimacy with her husband. Her husband has never abused her physically, psychologically, or sexually. Anna says she feels no stress about her husband’s problems. Neither depends on the other. The husband has a criminal record and drug problems.

Concerning her family of origin, her father is deceased, and her mother has psychiatric problems. Therefore, she and her siblings were placed in a children’s home during adolescence. She reports that she was sexually abused by her maternal uncle when she was three years old. Overall, the relationship with her siblings is good, although she reports that sometimes one of them criticises her severely and makes her feel useless. The data collected allows the attribution of the following scores for the FM subcomponent items (see Table 4).

Comparing the total score of FM with the risk/need levels defined by the authors of LS/CMI (see Table 2), we can say that family/marital represents a high-level risk factor for Anna.

#### 3.1.4. Leisure/Recreation (LR) Subcomponent of LS/CMI

Anna is not regularly involved in voluntary groups, and her free time is mainly spent reading. Based on these data, we attributed the following scores to the LR subcomponent items (see Table 5).

Comparing the total score of LR with the risk/need levels defined by the authors of LS/CMI (see Table 2), we can say that leisure/recreation represents a high-level risk factor for Anna.

#### 3.1.5. Companions (CO) Subcomponent of LS/CMI

Anna says that she and her husband lived in a poor neighbourhood where most people are convicted criminals. She knows many people who have trouble with the law, but she specifies that she also has old non-criminal friends who run businesses. Based on these data, we attributed the following scores to the CO subcomponent items (see Table 6).

Comparing the total score of CO with the risk/need levels defined by the authors of LS/CMI (see Table 2), we can say that companions represent a high-level risk factor for Anna.

#### 3.1.6. Alcohol/Drug Problem (ADP) Subcomponent of LS/CMI

Anna reports that she currently rarely drinks. In the past, she used cannabis (two joints a day) and alcohol but has not taken drugs for eleven years. She claims that alcohol and drug use may have contributed to her violations of the law. Her family was unaware of these habits, so she did not have any conflicts that could have led her to leave her home. She did not have problems at school or work because of drug or alcohol use; however, she reports that she became ill from drug and alcohol abuse. Based on these data, we attributed the following scores to the ADP subcomponent items (see Table 7).

Comparing the total score of ADP with the risk/need levels defined by the authors of LS/CMI (see Table 2), we can say that Alcohol/Drug Problem represents a medium-level risk factor for Anna.

#### 3.1.7. Procriminal Attitude/Orientation (PA) Subcomponent of LS/CMI

Concerning the crime (filicide), Anna admits to having made a serious mistake and thinks about the victim with sorrow. Nevertheless, she refers to it as an accident. She showed a strong detachment and was not moved, but she interrupted the interview to smoke a cigarette. She states that ‘obsessions contributed to the accident’. She would like to live her life without crime. She believes the law is fair, as her sentence and the courts who ruled on it. She also considers the security measures applied to her as fair, although she would like to be free. Based on these data, we attributed the following scores to the PA subcomponent items (see Table 8).

Comparing the total score of PA with the risk/need levels defined by the authors of LS/CMI (see Table 2), we can say that Procriminal Attitude/Orientation represents a very low-level (absent) risk factor for Anna.

#### 3.1.8. Antisocial Pattern (AP) Subcomponent of LS/CMI

Anna was assessed by several psychiatrists and psychologists who made different diagnoses: (1) schizoaffective disorder depressive type in a subject with borderline personality disorder; (2) peripartum depressive disorder; and (3) major depressive disorder with psychotic symptoms. She was never arrested before the age of 16, but she lived in a children’s home when she was a teenager. At 18 years old, she left home for a week. She was convicted of filicide before being acquitted because of being diagnosed as incompetent. She never escaped from an institutional setting, and she never received a suspension or revocation during the execution of the security measures. Regarding her economic situation, considering the different sources of income, she currently has no economic problems. Nevertheless, she says that she is worried about her car payments. She has a postal account for savings. During the last year, she never received any rejections for “insufficient funds.” She has no debts. During her life, she did not experience frequent changes in residence. She has been stably employed for more than four years. She attended high school until the second year without ever being suspended or expelled. Concerning the relationship with her parents, her mother has psychiatric problems (therefore, she was placed in a children’s home when she was a teenager), and her father is deceased. Anna defines the parental relationship as unrewarding. She does not have any specific hobbies; she spends her time working and reading. Concerning friendships, she has both friends with a criminal record and old non-criminal friends. The data collected allow the attribution of the following scores for the AP subcomponent items (see Table 9).

Comparing the total score of AP with the risk/need levels defined by the authors of LS/CMI (see Table 2), we can say that Antisocial Pattern represents a very low-level (absent) risk factor for Anna.

The total scores of the eight subcomponents of LS/CMI are summarised below (see Table 10).

In summary, for Anna, the LS/CMI administration showed that excluding the static (and therefore immutable) risk factor “criminal history” (CH = 2), the life areas that represent criminogenic risk factors/needs are:Family context (FM = 3), represented by the husband who still has problems with drug addiction and has a criminal record (although being supportive towards her); by the mother with psychiatric problems; and by a brother who sometimes assumed a judgmental attitude, making her feel useless. Through the administration of the tools, it was possible to learn about sexual abuse committed by her maternal uncle;The neighbourhood influences friendships (CO = 3) in which her family home is located (i.e., an area with a high crime rate). For this reason, Anna’s acquaintances and friendships mainly include people with criminal records. Nevertheless, her work allows her to broaden her knowledge and to also meet anti-criminal people;Leisure time (LR = 2). Although she participated in many activities in the past, she currently spends most of her time working, which does not give her much satisfaction, except from a strictly economic point of view. In her free time, she reads, an activity that does not stimulate the development of relationships;Alcohol/drug problems (ADP = 3). This area represents a medium risk factor for Anna. Although she has a significant history of alcohol and drug abuse (particularly cannabis)—she stated that she developed health problems related to it—she has been substance free for eleven years now.

Despite the risk/need factors listed above, the following aspects represent strengths for the recidivism’s prognosis for Anna:The absence of a pro-criminal attitude (PA = 0) corroborated because over ten years, and in different contexts, Anna has always shown awareness and shared the therapeutic objectives agreed with her;The absence of an antisocial pattern (AP = 0) confirmed by the diagnoses contained in the documentary consulted, her life history, and the reports produced over the years by the therapeutic residential community;Anna’s perseverance and commitment to work (EE = 1).

Overall, Anna was assessed at a medium recidivism risk level.

### 3.2. HCR-20 V3

#### 3.2.1. Historical Scale (H)

Anna did not commit violent crimes during childhood and adolescence. She committed filicide at the age of 32. In adulthood (24 years), she also committed a property crime (invasion of grounds and premises). Despite this, she did not have any other anti-social behaviour. She has a stable, non-violent intimate relationship with her partner. She is regularly married, and the relationship with her family (siblings) is good. She has good connections both with her work colleagues and with the guests and professionals of the therapeutic residential community where she currently lives. During the last ten years, interpersonal relationships have also improved. Before her current conviction, she worked occasionally (cleaning assistance) and was mainly a stay-at-home spouse. She dropped out of school in the second year of high school. Since her probation, she has had a steady job. She does not consider it satisfactory but necessary for her economic livelihood. She used cannabis continuously from 16 to 30 years old. At the time of the filicide, she was under the influence of cannabis. She reports that she also abused alcohol. To date, she has not used drugs for 11 years and rarely drinks alcohol. Concerning major mental disorders, she does not remember having any symptoms during adolescence. At the age of 30, she experienced psychotic symptoms, such as auditory hallucinations. There is also evidence of obsessions and worries about her daughters, such as an obsession with their well-being. She reports suicidal thoughts and attempted suicide in conjunction with filicide. Language and cognitive functions are preserved. She tends to withdraw socially because of a lack of confidence. Suspiciousness, paranoid thoughts, fear of judgement, and a sense of inadequacy were present. She feels she has been a victim, and she lacks security throughout her life: at the age of three, her uncle abused her. Her mother suffers from a psychiatric condition and has attempted suicide. She remembers the bizarre attitude of her mother. During her adolescence, she lived in a children’s home following her mother’s suicide attempt. She condemns violent behaviour. At the age of 30, she began psychopharmacological therapy: at first, she did not adhere to the treatment; today, she recognises the importance of pharmacological treatment. She considers the therapeutic residential community programme an experience that allows her to redeem her own mistakes; she has been supported, welcomed, understood, and not judged.

#### 3.2.2. Clinical Scale (C)

Anna reports being aware of her mental distress. She has learned to accept it and to treat it. She admits and is aware of her violent behaviour. She agrees with the need to take psychopharmacological therapy. She feels good to be emotionally supported by her family, and she is aware of monitoring actions on her path. However, her husband has drug addiction problems, for which he is followed by a local addiction service. No violent ideations or intentions are present. Delusions, hallucinations, suicidal ideation, and cognitive deterioration are absent. Currently, she feels calm and motivated, although her behaviour is restless. She does not describe herself as wicked. The good pharmacological compensation and positive therapeutic community programme allowed her stability and symptomatic compensation.

#### 3.2.3. Risk Management Scale (R)

The social services dealing with Anna are good; there are good relations and a good network for future projects; they are very present in project monitoring. The family home is stable. The environment is not excellent, but it is still suitable. The neighbourhood is considered “at risk.” Social network support is good. The treatment programme is a resource. There is no stress, and recovery time is respected.

Based on data collected through HCR-20 V3, we attributed the following scores to the items (see Table 11):

In summary—as shown in Table 11—on the first scale (historical—H) we found the following items: previous violence in adulthood (H1); the presence of other anti-social behaviour (H2) enacted when she was a young adult, probably because of the socioeconomic living conditions of that time, but which today do not represent a significant factor; prolonged inactivity at work (H4) which characterised the entire period prior to her current conviction, but which today does not represent a significant risk factor; the past abuse of alcohol and substances (H5), from which, however, she has been abstinent for more than eleven years; the manifestation of a major mental disorder (H6), which, to date, has been adequately compensated; the presence of some personality traits attributable to borderline personality disorder (H7), which, to date, are not reflected in any manifestation; and the presence of traumatic experiences (H8), not adequately processed and which still influence Anna’s life.

Concerning the second scale about the present risk factors (clinical—C), we found only a small amount of instability and behavioural restlessness, which have never evolved into violent or antisocial behaviour. These are probably attributable to Anna’s impatience towards the persistence of a custodial security measure, which requires her to reside in the therapeutic community. Finally, as also revealed by LS/CMI (subcomponent 3), from the future risk management items (scale R), it emerged that the personal support of Anna (R3) could be negatively influenced by the progress of her husband’s drug rehabilitation programme and the psychiatric condition of her mother.

## 4. Discussion

The aim of the present study was to assess the current recidivism risk and social dangerousness of an Italian filicidal woman to provide an example of a violence risk assessment procedure based on the use of LS/CMI tool while evaluating its suitability in the Italian context.

As demonstrated by previous research, when filicide occurs, assessment of recidivism risk becomes even more important, especially when the offender has other children. Structured interviews with offenders can be an important source of information. Considering the primary role of social stressors influencing the filicide phenomenon [11], structured interviews help guide professionals through the contextualisation of the event.

Although other researchers analysed the cases of filicidal mothers [59,60,61], few studies have focused on other aspects besides psychiatric issues. For example, Ben Ammar and colleagues [62] used the same research design as our study to analyse the association between Delusional misidentification syndromes (DMS) and the murder of a child by his mother. In an Italian study, Giacchetti and colleagues focused on personality traits in a sample of filicidal mothers, while other researchers focused on prevention measures [63].

In this study, data collected through LS/CMI and HCR-20 V3 revealed a range of information about different life areas that had not been underlined in the documentary survey. Knowing such information is useful both to define the persistence of offenders’ social dangers and to plan the offenders’ psychosocial rehabilitation programme. Although from the second year of Anna’s probation, the information about her condition increases, ambiguities, and inconsistencies between them increase as well. This is probably due to the different professionals who have dealt with this single case over the past 11 years. Ambiguities and inconsistencies are particularly evident in the determination of Anna’s psychopathological diagnosis.

We reported (see Box 0) that the diagnoses formulated during the Security Measure by different professionals both through the DSM-IV TR [64] and the DSM-5 [65] varied from schizoaffective disorder depressive type in a subject with borderline personality disorder to depressive disorder with psychotic symptoms to, finally, depressive disorder with post-partum onset specifier. However, it is well known that the diagnostic process operated through the DSM does not allow the above-mentioned diagnoses to be made simultaneously. This inconsistency hinders the objective process of Anna’s social dangerousness assessment so far based on the incurability of the disorder from which she suffers. In the context of this single case—since there is not even an unequivocal opinion as to the real disorder that Anna suffers from and the relative and potential prognosis—how can she be defined as incurable and then base the assessment of recidivism risk mainly on these prognoses?

As several authors argue [11,66], the hypothesis that Anna was affected peripartum by major depressive disorder with psychotic symptoms is well-suited to the situational picture presented. Taking this into account, the psychosocial rehabilitation programme of the offender deserves to be reviewed. The symptom compensation should be attributed not only to the psychopharmacological treatment but also to her containment in the therapeutic residential community. Rather, it could be traced back to stressful psychosocial factors, which gradually resolved and, consequently, contributed to the remission of her disorder [66,67,68].

Although the recidivism risk level cannot only be defined on the psychopathological condition of the offender, here, it is strongly relied upon to assert the persistence of social dangerousness. It seems like a diagnostic label was being used to fill the need for control.

This situation leads to a sort of endless security measure because, if the psychopathology is incurable—and containable through psychopharmacological treatment and therapeutic community containment—periodically checking the rehabilitation progress results is unnecessary. Continuing to restrict her autonomy hinders the achievement of a better balance between criminogenic needs and strengths and, therefore, her effective social reintegration.

It is more appropriate to consider possible relapses that may be prevented and managed when in the presence of strength elements, identified using scientifically validated assessment tools [66,67,68,69]. As shown by the results of this study, the use of LS/CMI seems to be appropriate for achieving this aim. It allows the assessment of standardised life areas, orienting the risk assessment process in a more reliable direction. Moreover, standardised tools facilitate communication between different professionals involved in the process, thus decreasing ambiguities due to autonomous practices. The discretionary nature of the assessment and the lack of specific tools could lead to a delay in the gradual acquisition of the autonomy level desired by the subject. In addition, the use of scientifically validated tools allows the rehabilitation programme to be planned in a way that is most useful for the user’s condition and to monitor improvements over time.

As specified by the risk principle of the Risk-Need-Responsivity model (RNR) [70], recovery, and social reintegration programmes should be calibrated according to the risk level of the individual. Intensive programmes should only target those who are at a moderate–high level, excluding those at a lower risk level, as it reduces contact with higher-risk offenders in criminal justice settings. The responsivity principle of the model also states that programmes should aim to maximise the competencies of the subjects and adapt to their learning style, degree of motivation and abilities.

In this single case, the LS/CMI showed that some sub-components could still represent a criminogenic need for her. To prevent anti-social behaviour, it would be necessary to take these data into account for more accurate planning of the subsequent phases of the psychosocial rehabilitation programme. Concerning family relations, it would be useful to promote a link between Anna’s services and that of her husband to preserve the supporting role that he provides over time and prevent it from becoming a risk factor for her.

Traumatic experiences lived within the family should be considered and consciously placed in Anna’s life story as part of a therapeutic process. The processing of traumatic experiences could take place in a peer support group in which Anna’s connection network can be expanded. In order to fill the lack of satisfaction and to gratify her work commitment, it would be appropriate to encourage Anna’s participation in other recreational activities—outside the community—more inclined to her interests. In this way, it would be possible to increase her motivation level and make these activities an external link and support for her.

Parallel to the criminogenic needs, LS/CMI highlighted Anna’s strengths: the absence of a pro-criminal attitude and an antisocial pattern, as well as her work commitment. We suggest leveraging this last aspect to keep her engaged and prevent possible clinical relapses. It could also be possible to increase the number of working hours or to propose a second job (the first one is a part-time job) more in line with her abilities and from which she can get greater satisfaction.

Actuarial tools do not allow for the identification of the strengths and weaknesses of the individual, which are useful for the development of treatment programmes suitable for reducing the risk of recidivism [43,70]. As demonstrated by this single case study, a violence risk assessment and risk management process guided by scientifically validated tools provides psycho-social data, which are fundamental to consider for prognosis, but which can be missed if professionals continue to use a purely clinical approach. Moreover, through the current study, we contributed to the adaptation of the Italian context to an international audience.

Future studies in this field should confirm the usefulness of this methodology and tools to systematise risk assessment and the risk management process and make it homogenous across countries.

The present study is not without some limitations. First, of the two instruments used in the evaluation activity, only the Italian validation study of HCR-20 V3 is currently available. Regarding LS/CMI, even though numerous studies have already demonstrated its effectiveness and reliability, the Italian validation is still in progress. This study is therefore the first exploratory contribution. Here, the HCR-20 V3 represented a support tool for the evaluation carried out using the Italian version of LS/CMI. The latter proved to have the application potential to solve the criticalities associated with complex clinical pictures, such as the one presented.

Second, it is well known that the most widely recognised limitation of single case studies concerns external validity. However, many researchers have stressed that this type of research best enhances the richness of clinical work with patients, without neglecting methodological aspects. If it is true that single cases are unique by their nature, it is also true that they always contain aspects that can be replicated [71]. The replicability of the study can be seen in the structure of the interview (e.g., LS/CMI- interview guide), which allows the same information to be collected for each administration. In this way, it is possible to obtain a standard assessment process for all the subjects examined. In fact, for the LS/CMI’s authors, the objective of well delineating the subject’s condition to support the prognostic judgement is fundamental.

## 5. Conclusions

In this work, we underlined how the LS/CMI assessment tool is useful in assessing the risk of reoffending and social dangerousness, and in identifying and managing social rehabilitation programmes.

As highlighted in the results, LS/CMI guides the interviewer through the exploration of eight areas of life considered risk factors and criminogenic needs for the offender. In this way, in addition to an overall recidivism risk score, we have been able to identify which areas need to be strengthened and which ones can be leveraged to make the social reintegration programme more effective. Despite the differences between the Canadian and Italian judicial systems, the tool has proved to be suitable in our context and useful for the evaluation of the participant.

From this single case emerged that although the evaluation carried out by HCR-20 V3 (which is a specific violence risk assessment tool already validated in Italy) provides useful information related to the risk of future violence, it is less detailed than LS/CMI collecting information about different life domains that can positively and negatively influence the social reintegration programme. Moreover—unlike HCR-20 V3—LS/CMI is provided as an interview guide. The possibility of using an interview guide adds more confidence to the interviewer and can make the evaluation faster.

Therefore, assessments guided by tools such as LS/CMI are even more valuable in emergencies—e.g., the last two years of the COVID-19 pandemic—where evaluations need to be carried out rapidly because rapid action needs to be taken in response to these assessments.

## Figures and Tables

**Table 1 ijerph-19-06967-t001:** Anna’s scores on the eight items of the subcomponent of Criminal History (CH).

Item	1	2	3	4	5	6	7	8	Total
Score	1	0	0	0	0	1	0	0	2

**Table 2 ijerph-19-06967-t002:** Risk/need profile of Level of Service/Case Management Inventory (LS/CMI).

Risk/Need	CH	EE	FM	LR	CO	ADP	PA	AP
Very high	8	8–9	4	-	4	7–8	4	4
High	6–7	6–7	3	2	3	5–6	3	3
Medium	4–5	4–5	2	1	2	3–4	2	2
Low	2–3	2–3	1	-	1	1–2	1	1
Very low	0–1	0–1	0	0	0	0	0	0

**Table 3 ijerph-19-06967-t003:** Anna’s scores on the nine items of the subcomponent Education/Employment (EE).

Item	9	10	11	12	13	14	15	16	17	Total
Score	0	0	0	0	1	0	0	0	0	1

**Table 4 ijerph-19-06967-t004:** Anna’s scores on the four items of the subcomponent Family/Marital (FM).

Item	18	19	20	21	Total
Score	0	1	1	1	3

**Table 5 ijerph-19-06967-t005:** Anna’s scores on the two items of the subcomponent Leisure/Recreation (LR).

Item	22	23	Total
Score	1	1	2

**Table 6 ijerph-19-06967-t006:** Anna’s scores on the four items of the subcomponent companions (CO).

Item	24	25	26	27	Total
Score	1	1	0	1	3

**Table 7 ijerph-19-06967-t007:** Anna’s scores on the eight items of the subcomponent Alcohol/Drug Problem (ADP).

Item	28	29	30	31	32	33	34	35	Total
Score	1	1	0	0	0	0	0	1	3

**Table 8 ijerph-19-06967-t008:** Anna’s scores on the four items of the subcomponent, Procriminal Attitude/Orientation (PA).

Item	36	37	38	39	Total
Score	0	0	0	0	0

**Table 9 ijerph-19-06967-t009:** Anna’s scores on the four items of the subcomponent Antisocial Pattern (AP).

Item	40	41	42	43	Total
Score	0	0	0	0	0

**Table 10 ijerph-19-06967-t010:** Total scores of the eight subcomponents of LS/CMI.

Subcomponent	CH	EE	FM	LR	CO	ADP	PA	AP	Total
Score	2	1	3	2	3	3	0	0	14

**Table 11 ijerph-19-06967-t011:** Scores attributed to the items of the Historical Clinical Risk 20–Version 3 (HCR-20 V3).

Items	Presence	Relevance
Historical (H)		
H1 Previous violence	Yes	Medium
H2 other antisocial behaviour	Partial	Low
H3 relationships instability	No	- ^1^
H4 employment problems	Partial	Low
H5 substance use problems	Partial	Low
H6 major mental illness	Yes	Low
H7 personality disorders	Partial	Low
H8 traumatic experiences	Yes	Medium
H9 violent attitudes	No	-
H10 treatment or supervision response	No	-
Clinical (C)		
C1 insight	No	-
C2 violent ideation or intent	No	-
C3 symptoms of major mental disorder	No	-
C4 instability	Partial	Low
C5 treatment or supervision response	No	-
Risk Management (R)		
R1 professional services and plans	No	-
R2 living situation	No	-
R3 personal support	Partial	Low
R4 treatment or supervision response	No	-
R5 stress or coping	No	-

^1^ not applicable.

## Data Availability

Data are available upon request.

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
