# Peer review of "Violence Risk Assessment and Risk Management: Case-Study of Filicide in an Italian Woman"

_ijerph, 2022, doi:10.3390/ijerph19126967_

Round 1
Reviewer 1 Report
- Usually keywords don't take (over) multiple sequences from the title - I recommend replacing them so that they can reflect the ideas in the article and not just be redundant
- I consider that the contents of Box 1 should be placed at the end of the article in Annex 1
- I have not identified any individualized / separate area for Literature Review - I think it should be completed
- I consider that there should be a (sub)chapter stating "clearly" the Results of the research (eg: the "measurable" impact on the phenomenon, at national / international level) or some phrases
- It would be appreciated if the authors would introduce a subchapter / chapter in which they would propose some "concrete" measures of alignment and convergence of measures for a somehow unitary functioning.
Author Response
Dear Reviewer,
Thank you for taking the time to review our work and for giving us further ideas to improve our work. We have been able to incorporate changes to reflect your suggestions. We have highlighted the changes within the manuscript.
- Usually keywords don't take (over) multiple sequences from the title - I recommend replacing them so that they can reflect the ideas in the article and not just be redundant
- I consider that the contents of Box 1 should be placed at the end of the article in Annex 1
- I have not identified any individualized / separate area for Literature Review - I think it should be completed
- I consider that there should be a (sub)chapter stating "clearly" the Results of the research (eg: the "measurable" impact on the phenomenon, at national / international level) or some phrases
- It would be appreciated if the authors would introduce a subchapter / chapter in which they would propose some "concrete" measures of alignment and convergence of measures for a somehow unitary functioning
Thank you for your comments. We substituted some of the keywords in order to better reflect the idea and the target of our work. As you suggested we placed Box1 in Appendix A. Moreover, we put the current situation of the case at the beginning of the case report in order to meet the suggestion of another reviewer. As regards to the literature review, as stated in Figure 1, we focused on filicide phenomenon, offenders’ characteristics, filicide risk factors and on violence risk assessment tools. As you suggested we added data about the impact of the phenomenon at international and national level. Unfortunately, we were not able to fully understand the meaning of your latest comment. If you would like to suggest the proposal of another assessment tool in order to measure the functioning of the case study patient highlighting elements of convergence and divergence with LS/CMI tool, we tried to do it through the HCR-20 V3 tool (already validated in Italy). We apologize if we did not understand the meaning of your suggestion.
Thank you
Kind regards
Reviewer 2 Report
- You should check the Vancouver guidelines in the bibliography of the text. For example, when it says 21, 22, 23, 24, 25, 26, the correct text is 21-26.
- The assessment tools should be described in the material and methods section and not in the introduction, as they appear in this manuscript.
- No research objective has been raised in this manuscript and it is necessary
- In this study, no design has been proposed and should be carried out
- In the discussion of a research study, similarities and differences with studies by other authors should be raised, but assumptions should not be questioned, for these matters there is a section on future lines of research.
- The limitations of the study should not be stated in the conclusions, they should be described in the specific section for this purpose.
- Future lines of research should not be considered in the conclusions section either, there is a specific one for this.
- There are some very old bibliographic references that need to be modified by more current ones.
Author Response
Dear Reviewer,
Thank you for taking the time to review our work and for giving us further ideas to improve our work. We have been able to incorporate changes to reflect your suggestions. We have highlighted the changes within the manuscript.
- You should check the Vancouver guidelines in the bibliography of the text. For example, when it says 21, 22, 23, 24, 25, 26, the correct text is 21-26.
- The assessment tools should be described in the material and methods section and not in the introduction, as they appear in this manuscript.
- No research objective has been raised in this manuscript and it is necessary
- In this study, no design has been proposed and should be carried out
- In the discussion of a research study, similarities and differences with studies by other authors should be raised, but assumptions should not be questioned, for these matters there is a section on future lines of research.
- The limitations of the study should not be stated in the conclusions, they should be described in the specific section for this purpose.
- Future lines of research should not be considered in the conclusions section either, there is a specific one for this.
- There are some very old bibliographic references that need to be modified by more current ones.
Thank you for your comments. We adapted all the references in the text and the bibliography to the MDPI style.
As you suggested, we placed the tools’ description in the material and method section. We stated our objectives clearer in the abstract, in the introduction section and in the discussion. We indicated the study design in the procedure subchapter (material and method section) and in the Figure 1. As you suggested we better organized our content in the discussion section including future lines and limitations of our study in line with the template provided by the journal. As regards to the old references, indeed some are very old, but refer to the validation study of some instruments (e.g., Rorschach) or to important studies for the target phenomenon. In these cases, we chose to keep them, but we also added newest references.
Thank you
Kind regards
Reviewer 3 Report
I consider the primary critical need in this paper is the bibliographic reference review as a whole. The references should be reviewed, following the authors' instructions in the journal's regulations. There are several inconsistencies according to the Vancouver Standards.
Please, consult the link below.
https://www.mdpi.com/journal/ijerph/instructions
https://www.mdpi.com/journal/ijerph/instructions
This paper has the characteristic that the sample is limited to one subject. Still the authors have built the aim of the study to highlight the relevance of more objectives and analytical tools to help forensic decisions, which is a relevant strength.
​On the other hand, authors should reorder the place for limitations (line 490) and future studies (line 516). They must be in the discussion and not in conclusions.
Author Response
Dear Reviewer,
Thank you for taking the time to review our work and for giving us further ideas to improve our work. We have been able to incorporate changes to reflect your suggestions. We have highlighted the changes within the manuscript.
I consider the primary critical need in this paper is the bibliographic reference review as a whole. The references should be reviewed, following the authors' instructions in the journal's regulations. There are several inconsistencies according to the Vancouver Standards. Please, consult the link below
https://www.mdpi.com/journal/ijerph/instructions
This paper has the characteristic that the sample is limited to one subject. Still the authors have built the aim of the study to highlight the relevance of more objectives and analytical tools to help forensic decisions, which is a relevant strength. On the other hand, authors should reorder the place for limitations (line 490) and future studies (line 516). They must be in the discussion and not in conclusions.
Thank you for your comments. We adapted all the references in the text and the bibliography to the MDPI style. As you suggested, we better organized our content in the discussion section including future lines and limitations of our study in line with the template provided by the journal.
Thank you
Kind regards
Reviewer 4 Report
Dear Authors
Very interesting article about the filicide and it is also a burning and topical issue, but you have to pay attention to some points.
Firstly, you have to state the purpose of the study in the abstract and in the text
Figure 1 could be presented in another way?
A lot of important information is provided which unfortunately is not legible in the figure
How did this research start?Was it part of a larger study or was it made specifically for this incident?
In addition, the reason why the case is presented in context is not understood.
In the description of the case the current situation must be presented first and then the personal history.
Psychometric tools are over-analyzed, no needed.Also, the results are presented in great detail
Was there consent of the patient or someone else to participate in the study and the publication?
What was so special about this case to be a case study, since the woman has a psychiatric background and as an incident unfortunately, it is not isolated. Please highlight the answer in the text.
The section stretches and limitations should be mentioned separately because it does not belong to the conclusions.
You do not state your conclusions in this section
Author Response
Dear Reviewer,
Thank you for taking the time to review our work and for giving us further ideas to improve our work. We have been able to incorporate changes to reflect your suggestions. We have highlighted the changes within the manuscript.
Dear Authors,
Very interesting article about the filicide and it is also a burning and topical issue, but you have to pay attention to some points.
Firstly, you have to state the purpose of the study in the abstract and in the text
Figure 1 could be presented in another way? A lot of important information is provided which unfortunately is not legible in the figure.
How did this research start? Was it part of a larger study or was it made specifically for this incident?
In addition, the reason why the case is presented in context is not understood.
In the description of the case the current situation must be presented first and then the personal history.
Psychometric tools are over-analyzed, no needed. Also, the results are presented in great detail
Was there consent of the patient or someone else to participate in the study and the publication?
What was so special about this case to be a case study, since the woman has a psychiatric background and as an incident unfortunately, it is not isolated. Please highlight the answer in the text.
The section stretches and limitations should be mentioned separately because it does not belong to the conclusions.
You do not state your conclusions in this section
Thank you for your comments. We stated our objectives clearer in the abstract, in the introduction section and in the discussion. As you suggested, we changed the shape of the Figure 1 and we placed it in the Appendix A. We hope that it could be clearer now. We stated how our research start and further information both at the end of the introduction section and in the procedure subchapter (material and method section). We placed the Box 1 in the Appendix A and we put the current situation of the case at the beginning of the case report. As you suggested we shortened the description of psychometric tools. As regard to the informed consent, we did not include it because of the mandatory nature of our evaluations authorized both by the case’s supervising judge and the head of the residential therapeutic community. As you suggested, we stated it in the procedure subchapter. In conclusion, we better organized our content in the discussion section including future lines and limitations of the study in line with the template provided by the journal. Moreover, we extended the content of our conclusions.
Thank you
Kind regards
Round 2
Reviewer 4 Report
The most issues have been adressed. However, you should rename the paper as case study.
Author Response
Dear Reviewer,
Thank you for taking the time to review our work. Thank you for your latest comment and for giving us further suggestion to improve our work.
As you suggested, we had the article proofread by a native English-speaking professor.
Kind regards